# Experimental Study on Ultrasonic Vibration-Assisted WECDM of Glass Microstructures with a High Aspect Ratio

**DOI:** 10.3390/mi12020125

**Published:** 2021-01-26

**Authors:** Yan Chen, Xu Feng, Gongming Xin

**Affiliations:** School of Energy and Power Engineering, Shandong University, Jinan 250061, China; chenyan79@sdu.edu.cn (Y.C.); fxwd148397mr@163.com (X.F.)

**Keywords:** ultrasonic vibration, helical electrode, WECDM, high aspect ratio, glass microstructures

## Abstract

With the rapid development of micro-electro-mechanical systems (MEMSs), the demand for glass microstructure is increasing. For the purpose of achieving high quality and stable machining of glass microstructures with a high aspect ratio, ultrasonic vibration is applied into the micro-wire electrochemical discharge machining (WECDM), which is proposed as ultrasonic vibration-assisted WECDM with a micro helical electrode. Firstly, the formation of a gas film on the surface of the helical electrode in WECDM machining is simulated, meaning the thickness of the gas film can be reduced by adding suitable ultrasonic amplitude, thus reducing the critical voltage, then the machining localization and stability were enhanced. Then, the micro helical electrode with a diameter of 100 μm is used to carry out sets of experiments that study the influence of ultrasonic amplitude, machining voltage, duty factor, pulse frequency, and feed rate on the slit width. The experimental results show that the machining stability and quality are significantly improved by adding suitable ultrasonic amplitude. When the amplitude was 5.25 μm, the average slit width was reduced to 128.63 μm with a decrease of 20.78%. Finally, with the optimized machining parameters, micro planar coil structure and microcantilever structure with a high aspect ratio were fabricated successfully on the glass plate. It is proved that ultrasonic vibration-assisted WECDM with the micro helical electrode method can meet the requirements of high aspect ratio microstructure machining for hard and brittle materials.

## 1. Introduction

With the rapid development of micro-electro-mechanical systems (MEMSs), the demand for glass microstructures is increasing, due to their excellent properties. However, because of the hardness, brittleness, and non-conductive properties of glass, it is very difficult to machining complex microstructure [1,2,3]. Wire electrochemical discharge machining (WECDM) is a kind of non-contact machining, which can overcome its hard brittleness and non-conductive characteristics, so it can be competent for the machining of glass, a kind of non-conductive hard, brittle insulating material [4,5].

In recent years, many scholars have been actively engaged in the field of electrochemical discharge machining (ECDM) and achieved certain results. Kulkarni [6] discussed the mechanism of temperature rise and material removal by observing and recording the current signal. Kim et al. [7] studied the influence of pulse frequency and duty factor on glass machining by ECDM using a pulse power supply. The experimental results showed that the thermal damage diminishes with the rise of pulse frequency and the decrease of duty factor. Nguyen et al. [8] studied the microstructure of ECDM ceramics and discussed the influences of electrolyte level and concentration, pulse voltage, pulse opening and closing time, and tool feed rate on the machining performance. Hajian [9] studied the effects of magnetic field direction and electrolyte concentration on the microstructure of glass in ECDM, and the effects of machining voltage, field orientation, and electrolyte concentration on the machining performance. It was found that the magnetic field orientation affected the bubble movement direction. Liu et al. [10] established the theoretical model of electrochemical drilling (ECD) with ultra-short voltage pulse, analyzed the influence of the helical electrode rotation on the gap flow field. Elhami and Razfar [11] used ultrasonic vibration tool electrode to study ECDM, established a mathematical model, and analyzed the influence factors of gas film thickness. Goud et al. [12] reviewed and listed the methods to improve the material removal rate in ECDM. Liu et al. [13] analyzed the influence of high-speed rotation of electrodes on promoting electrolyte and preventing secondary electrolysis, and proved that high-speed rotation of electrodes can enormously raise machining efficiency and constancy. Xiong [14] proposed a new method of rotary electrochemical etching and introduced the mechanism of rotary electrochemical etching with a micro helical cylindrical electrode. Zhang [15] studied the concentration distribution of products machined by tube electrode tools with the spiral structure in the machining gap, indicating that trapezoidal spiral structure is conducive to effectively removing products from the machining gap, to enhance the removal effect of machined products.

As a branch of ECDM, WECDM has a great influence on the machining of hard and brittle insulating materials, and many scholars have carried out in-depth research on it. Sunder [16] summarized the research status of WECDM in the world, and optimized the machining parameters of WECDM. Malik et al. [17] used brass wire with a diameter of 200 μm to cut the epoxy glass fiber composite. The machined slit width can reach 220 to 223 μm. Wang [18] used WECDM-assisted diamond wire cutting of glass. Ankit [19] used coated steel wire with a diameter of 0.15 mm for the first time to study quartz cutting. The effects of applied voltage, electrolyte concentration, and line speed on the material removal rate and slit width were analyzed. Pallvita Yadav [20] studied the cutting of silica epoxy nanocomposites, and discussed the effects of voltage, concentration, wire-speed, pulse width, and silicon particle concentration on the material removal rate and surface roughness.

Many scholars have made great achievements in the research of ECDM and WECDM, but there is no research on the application of ultrasonic vibration to WECDM. Han et al. [21] applied ultrasonic vibration to the electrolyte to quickly update the electrolyte circulation during drilling, to improve the drilling depth and inlet quality. Rusli M [22] introduced ultrasonic vibration assistance into ECDM drilling to improve surface quality and material removal rate. Ultrasonic vibration can bring huge gain effect for the quality of WECDM, so this paper introduces ultrasonic vibration into the micro-WECDM, and innovatively puts forward the micro helical electrode ultrasonic vibration-assisted WECDM method, to solve the problem of machining glass complex micro planar coil structure and high aspect ratio microcantilever structure.

Firstly, the formation of a gas film on the surface of the helical electrode in the WECDM process was simulated, meaning the thickness of the gas film can be reduced by adding suitable ultrasonic amplitude, thus reducing the critical voltage, then the machining localization and stability were enhanced. Then, the micro helical electrode with a diameter of 100 μm was used to carry out sets of experiments to study the influence of ultrasonic amplitude, machining voltage, duty factor, pulse frequency, and feed rate on the slit width. The experimental results show that the machining stability and quality were significantly improved by adding suitable ultrasonic amplitude. When the amplitude was 5.25 μm, the average slit width was reduced to 128.63 μm with a decrease of 20.78%. Finally, with the optimized machining parameters, micro planar coil structure and microcantilever structure with a high aspect ratio were fabricated successfully on the glass plate.

## 2. Principle and Experimental Set-Up

### 2.1. The Machining Principle

According to the basic principle of WECDM, it is necessary to produce small bubbles on the tool electrode, and then polymerize them into an insulating gas film to isolate the tool electrode from the electrolyte. Finally, the surface material of the workpiece can be removed by the energy generated by the discharge, as shown in Figure 1. 

The specific machining circuit is shown in Figure 2a. R1 is the resistance at the junction of the tool electrode and electrolyte. R2 is the resistance at the junction of the auxiliary anode and electrolyte. R3 is electrolyte resistance. L is the loop inductive reactance. C1 is the capacitance at the tool electrode and electrolyte. C2 is the capacitance at the auxiliary anode and electrolyte. In the actual process of machining, whether the resistance or capacitance, or inductance is dynamic, which is one of the reasons for the current fluctuation in the loop. When the small bubbles attached to the tool electrode gradually form an insulating gas film to wrap the tool electrode, the current on the electrode drops sharply to zero, which can be understood as the R1 resistance is infinite. When the voltage rises above the critical value, the gas film will be broken down instantaneously to produce stable discharge. At this time, R1 can be understood as the resistance value is zero. In addition, the values of R2 and R3 will also change with the altering of local temperature of bubbles and electrolyte, and the capacitance will change with the gap.

According to the different voltage applied on the tool electrode and auxiliary anode, the voltage current characteristic curve of electrochemical ECDM is obtained, as shown in Figure 2b. The voltage current static characteristic curve can be divided into five parts:Cut off zone: When the voltage is less than *U_d_*, there is no electrolytic reaction.Linear region: The voltage and current in section *AB* are approximately linear. An electrolytic reaction takes place at that time, and the current increases linearly with the voltage.Saturation region: The voltage in *BC* section increases slowly with the current and gradually reaches saturation value.Jump region: With the increase of voltage in the *CD* section, the gas film formed gradually envelops the electrode, and the current decreases rapidly.Discharge area: When the voltage in *DE* section exceeds the critical value of *U^crit^*, the breakdown gas film is discharged stably, and the surface material of the workpiece is removed.

It can be seen from Figure 2b when the voltage exceeds the critical voltage *U^crit^*, the machining can change from electrolysis to ECDM. In this paper, the critical voltage from electrolytic jump to ECDM is 23 V, and the current waveform is shown in Figure 3. The corresponding current waveform in every single pulse is as follows: At the beginning of the loading voltage, the tool electrode contacts the electrolyte in a large area for electrolysis, and the current increases instantaneously. With the increase of the number of small bubbles, the tool electrode is gradually isolated from the electrolyte, and the current drops rapidly close to zero. When the bubble film is broken down, the current rises instantaneously, at the same time, the electrode surface contacts with the electrolyte again for electrolysis, and the generated bubbles form a gas film to isolate the tool electrode again and start the next discharge.

Voltage is the key factor of ECDM, and the critical voltage is a characterization of whether it is easy to carry out. The magnitude of critical voltage is related to many factors, mainly to the impedance in the circuit. In this paper, ultrasonic vibration-assisted WECDM is used. Through many experiments, it is found that the ultrasonic amplitude has a great influence on the critical voltage. Ultrasonic vibration is applied to the tool electrode to do axial vibration, which can refine the gas film and make it thinner and more uniform, thus reducing the critical voltage of discharge.

### 2.2. The Machining Set-Up

In this paper, ultrasonic vibration is introduced into micro-WECDM, and the experimental platform has the characteristics of high motion control accuracy, antivibration and high stability. The physical picture of the experimental platform is shown in Figure 4. The experimental platform mainly consists of the following parts: Machine body, ultrasonic generator, and ultrasonic motorized spindle, pulse power supply, frequency converter, electrode system, monitoring, and testing system.

The base of the machine is made of granite, which can absorb vibration and resist corrosion. The motion accuracy of the machine tool can reach 0.1 μm/step, which can meet the precision requirements of micromachining. The ultrasonic vibration comes from the ultrasonic generator, the vibration frequency is between 20,000~31,000 Hz, and the amplitude is continuously adjustable. The vibration amplitude can be measured and controlled by the ultrasonic generator. The power supply is an important energy control terminal when cutting. This paper uses a rectangular wave pulse power supply. The output energy is controlled by controlling the pulse voltage, frequency, and duty factor. In order to control the spindle speed accurately, Delta VFD-B AC frequency converter is used to control the machining speed of the helical electrode. The electrode system mainly includes a tool electrode (helical electrode with a diameter of 0.1 mm and tungsten wire with a diameter of 0.05 mm), electrolyzer, auxiliary anode graphite, manual lifting table, and auxiliary fixture. In order to observe the machining state of cutting, the current signal is captured by Hall current sensor in the circuit and observed online through an oscilloscope. To record the current signal during machining, the current sensor and Advantech usb4711A data acquisition card are used in the experiment. To measure and observe the morphology and size after machining, an electron scanning microscope and laser confocal microscope were used.

## 3. Analysis of Gas Film Forming

The formation of a gas film is a very important process for ECDM. Hydrogen bubbles produced by electrolysis accumulate, combine and grow on the surface of the cathode tool, and finally form bubble film. There is also a view that the high temperature after the discharge of the tool electrode will also make the electrolyte vaporize to form bubbles. This paper mainly discusses the gas film formed by hydrogen bubbles produced by electrolysis. The quality and thickness of the gas film have a great impact on the quality and material removal rate of micro ECDM. A stable gas film can also improve the repeatability of ECDM.

For the purpose of studying the beneficial influence of ultrasonic vibration on the tool electrode, the Fluent software under ANSYS is used to simulate the impact of ultrasonic vibration on the gas film. Before simulating the impact of ultrasonic vibration on the gas film on the electrode, the model is simplified as follows: The electrolyte is set as an incompressible Newtonian fluid, the concentration and viscosity coefficient of the fluid remain unchanged, and the hydrogen bubble generated by electrolysis is replaced by velocity outlet.

The helical electrode is selected as the tool electrode. Due to the comparison of the influence of ultrasound on the gas film, the model can be in the non-machining state, and its geometric model is shown in Figure 5. The appearance of the model is a cuboid with the size of 1 mm × 1 mm × 0.8 mm, and the shape of the helical electrode is removed in the middle. The upper surface marked green (B1) is set as the pressure outlet, which is the interface between electrolyte and air. The diameter of the helical electrode marked with B2 is 0.1 mm, and the immersion depth is 0.6 mm. It is set as the velocity outlet of hydrogen to simulate the bubble generated by electrolysis. The four vertical surfaces and the bottom surface of the model (B3, B4, B5, B6, and B7) are set as the wall without sliding, and the solid part of the model is set as 3 mol/L KOH solution.

After the grid is divided, the multiphase flow model is selected, and the liquid is set as the main phase and the gas as the second phase. The k-ε model was selected according to the experimental conditions and the outlet velocity of hydrogen. The model consists of turbulent kinetic energy equation *k* and diffusion equation *ε* is as follows:(1)$$\frac{\partial }{\partial t}\left(\rho_{k}\right)+\frac{\partial }{\partial x_{i}}\left(\rho_{k}\mu_{i}\right)=\frac{\partial }{\partial x_{j}}\left[\left(\mu +\frac{\mu_{i}}{\sigma_{k}}\right)\frac{\partial k}{\partial x_{j}}\right]+G_{k}+G_{b}-\rho_{\varepsilon }-Y_{m}+S_{k}$$
(2)$$\frac{\partial }{\partial t}\left(\rho_{\varepsilon }\right)+\frac{\partial }{\partial x_{j}}\left(\rho_{\varepsilon }\mu_{i}\right)=\frac{\partial }{\partial x_{j}}\left[\left(\mu +\frac{\mu_{i}}{\sigma_{\varepsilon }}\right)\frac{\partial_{\varepsilon }}{\partial x_{j}}\right]+G_{1\varepsilon }\frac{\varepsilon }{k}\left(G_{k}+C_{3\varepsilon }G_{b}\right)-C_{2\varepsilon }\rho \frac{\varepsilon^{2}}{k}+S_{\varepsilon \;\;\;\;\;\;\;\;}$$
where *G_k_* is turbulent kinetic energy with laminar velocity gradient, *G_b_* is turbulent kinetic energy produced by buoyancy, *Y_m_* is fluctuation caused by excessive diffusion. *Δ_k_* is the turbulent Prandtl number of k equation, *δ_ε_* is the turbulent Prandtl number of the *ε* equation, *C*_1_, *C*_2_, and *C*_3_ are constants.

In the parameter setting, the diameter of hydrogen bubble is set as 2 μm, and the average outlet velocity is 0.0005 m/s according to the experimental conditions. 3 mol/L KOH and 1140 kg/m^3^ density were used. The vibration of the helical electrode boundary is controlled by the UDF. The displacement function is as follows:(3)$$y\left(t\right)=-A\cos\frac{2\pi t}{T}$$
where *A* is the amplitude and *T* is the period. For the purpose of simulating the influence of ultrasonic vibration on the gas film, the amplitude of 4 μm and the corresponding frequency of 25,000 Hz are selected. In the simulation, the time step is set at 1 μs. The simulation results are shown in Figure 6. Blue represents electrolyte, red represents hydrogen, and color depth represents corresponding volume fraction. The diameter of the electrode is 100 μm, and the gas film is dark red.

## 4. Result and Discussion

In this paper, the helical electrode was used to carry out the experimental research on the WECDM of glass, assisted by ultrasonic vibration. Then, the influence of ultrasonic vibration on the cutting slit width is mainly studied. The influence of ultrasonic amplitude, machining voltage, duty factor, pulse frequency, and feed rate on the cutting slit width is analyzed. Finally, the cutting machining of glass complex micro planar coil structure and microcantilever beam structure with high depth diameter ratio is completed by using the optimized machining parameters. The range of fixed parameters, such as spindle speed, electrolyte concentration, and other machining parameters, are shown in Table 1.

### 4.1. Effect of Ultrasonic Amplitude on Slit Width

The difference of machining effect with or without ultrasonic vibration assistance is shown in Figure 7. With the assistance of ultrasonic vibration, the average width of the entrance and exit decreases from 143.1 μm to 130.77 μm, and 137.3 μm to 128.17 μm, respectively. Through comparison, it is found that the width of the micro slit can be effectively reduced after ultrasonic vibration.

Based on the beneficial effect of ultrasonic vibration, the influence law of different amplitude on slit width is studied. In the experiment, the single variable method is used, other machining parameters remain unchanged, only the amplitude is changed. The machining voltage is 35 V, the duty factor is 70%, the frequency is 1000 Hz, the feed rate is 1 μm/s, and the spindle speed is 3000 r/min. The maximum amplitude is 7 μm. The average slit width under different amplitudes is shown in Figure 8. In Figure 8, the average slit width gradually decreases from 162.38 μm to 128.63 μm when the ultrasonic vibration is not applied until the amplitude increases to 5.25 μm. When the amplitude exceeds 5.25 μm and increases to 7 μm, the slit width increases to 161.68 μm. This shows that in the small amplitude range, ultrasonic vibration can reduce the slit width after cutting, and when the amplitude exceeds the critical value, it will increase the slit width.

Through the influence of different amplitude on the slit width after cutting, it can be found that the ultrasonic amplitude is not the greater, the better, but can improve the machining quality in a matching range. To further explore how the ultrasonic amplitude affects the machining quality, the current waveforms in the machining circuit with different amplitudes are recorded by an oscilloscope, as shown in Figure 9. The current signal in machining is analyzed below.

It is found that the number of single pulse spark discharge is much lower than that of ultrasonic-assisted spark when the amplitude is 0. Moreover, the peak current of spark discharge without ultrasonic vibration is higher than that without ultrasonic vibration. After applying ultrasonic vibration to the tool electrode, the gas film can be refined to make it thin and uniform. As the thickness of the gas film decreases, the critical voltage of the breakdown film also decreases, so the peak current of the spark discharge also decreases. The better film state makes the discharge stable and higher frequency. Finally, the slit with a smaller width and higher dimensional accuracy is machined.

When the amplitude increases from 1.75 μm to 5.25 μm, the peak current decreases gradually. With the increase of amplitude, the film thickness decreases. The corresponding critical discharge voltage and the peak current at discharge decrease. Therefore, in the small energy and high-density spark discharge, the cutting slit width is reduced. However, when the amplitude increases from 5.25 μm to 7 μm, the slit width increases. As shown in Figure 9, in the current waveforms with amplitudes of 5.25 μm and 7 μm, it can be found that with the increase of amplitude, the formation of the gas film will deteriorate, resulting in an increase in the time for the first formation of the gas film. Through the experimental analysis, when the amplitude is large, the gas film will be pulled violently, which makes the distribution of the gas film more uneven. Due to the uneven distribution of the gas film under large amplitude, the critical voltage of spark discharge at the electrode surface is high or low, and the peak current is uneven in the current waveform. When the discharge current is too high, the material removal energy will increase.

### 4.2. Effect of Machining Voltage on Slit Width

Machining voltage plays a very important role in WECDM. On the one hand, it is necessary to supply pressure to a large number of bubbles generated by electrolysis; on the other hand, it is necessary to supply voltage for spark discharge generated by the breakdown of the gas film. For the purpose of studying the influence of pulse voltage on the slit width after cutting, the single variable method is used to carry out the experimental research. In the experiment, all the workpieces are made of quartz glass with a thickness of 300 μm, the discussion range of pulse voltage is 33 V to 38 V, the tool electrode amplitude is 5.25 μm, the duty factor is 70%, the frequency is 1000 Hz, the spindle speed is 3000 rpm, and the feed rate is 1 μm/s.

The average slit width under different machining voltages is shown in Figure 10. There are two groups in the experiment, one is with ultrasonic vibration assistance, and the other is without ultrasonic assistance. The average slit width increases with the increase of voltage, whether or not there is ultrasonic assistance. The results show that the average slit width can be reduced after applying ultrasonic vibration to the tool electrode; the reduction is close to 10 μm. In addition, the minimum machinable voltage without ultrasonic vibration is 34 V, while the minimum machinable voltage with ultrasonic vibration is reduced by 1 V. With the help of ultrasonic vibration, the critical value of WECDM can be reduced. Although the minimum machinable voltage is reduced by only 1 V, the machining quality will be improved if the material is removed with less discharge energy.

### 4.3. Effect of Duty Factor on Slit Width

The material removal of WECDM mainly depends on the instantaneous heat during discharge, and the main source of this heat is the output voltage, duty factor, and frequency of the power supply. For the purpose of studying the influence of the duty factor on the slit width, experimental studies were carried out on a 300 μm thickness glass, at a constant frequency of 1000 Hz. The duty factor decreases from 90% to the minimum value of 50%. The average slit width and current waveform after cutting under different duty factors are shown in Figure 11 and Figure 12, respectively.

With the increase of duty factor, it will provide favorable conditions for the formation of a gas film and more spark discharges, so the discharge times in a single pulse increase significantly. With the increase of single pulse discharge times, the spark discharge energy increases correspondingly, and the gap width also increases. Cutting glass is mainly to remove heat during discharge, and the moment of heat generated by spark discharge is extremely short. In such a short time, the heat will be dissipated in the surrounding electrolyte, and the heat transferred to the glass surface is very limited. Therefore, in a single pulse, enough discharge times are needed to continuously transfer the heat to the workpiece. Therefore, there is a minimum duty factor in the cutting machining. If the duty factor is less than 50% in this experiment, the tool will be broken directly.

Considering the influence of duty factor on the slit width, it can be found that the duty factor should not be too high, otherwise the slit width will increase. However, the duty factor should not be too low, otherwise it cannot accumulate enough heat to continuously remove the workpiece material, and even cause tool electrode fracture. To sum up, choosing the appropriate duty factor and ensuring that the tool electrode does not collide with the workpiece, and the micro removal of workpiece material with small discharge energy will greatly improve the machining quality.

### 4.4. Effect of Pulse Frequency on Slit Width

Through a large number of experiments, it is found that the influence of duty factor and pulse frequency on machining quality is realized by controlling the pulse width and length of a single pulse. For the purpose of exploring the effect of pulse frequency, the duty factor is fixed in the experiment, and the value is 70%. The glass thickness is 300 μm, the amplitude is 5.25 μm, the voltage is 35 V, the feed rate is 1 μm/s, and the spindle speed is 3000 rpm. Considering that the smaller the frequency is, the larger the corresponding pulse period is, and the easier the machining is. Therefore, the frequency discussion range starts from 500 Hz, and the interval increases by 500 until it reaches 2500 Hz, and the normal cutting cannot be carried out. According to the experimental study, the average slit width at different frequencies is shown in Figure 13. The average slit width decreases with the increase of frequency, and the maximum machining stability is obtained at 1000 Hz.

### 4.5. Effect of Feed Rate on Slit Width

For the purpose of exploring the influence of feed rate on the slit width under the assistance of ultrasonic vibration, different comparative experiments were carried out on 300 μm thickness quartz glass. Considering that the lower the feed rate is, the stronger the cutting ability is. Therefore, the minimum feed rate starts from 0.5 μm/s and increases every 0.5 until it cannot be machined. In the experiment, the voltage is 35 V, the duty factor is 70%, the frequency is 1000 Hz, the amplitude is 5.25 μm, and the spindle speed is 3000 rpm. The average slit width at different feed rates is shown in Figure 14. At the same feed rate, the slit width with ultrasonic assistance is lower than that without ultrasonic assistance, which proves again that applying ultrasonic assistance on the tool electrode can effectively reduce and improve the dimensional accuracy of cutting. It can also be concluded from Figure 14 that the average slit width decreases with the rise of the feed rate, and the higher the feed rate is, the more rapid the slit width decreases. In this experiment, the maximum feed rate without ultrasonic assistance is 2 μm/s, while the maximum feed rate with ultrasonic assistance is increased to 2.5 μm/s. Compared with no ultrasonic assistance, the maximum feed rate with ultrasonic assistance is increased by 25%. In the experiment, a constant feed rate is used, so applying ultrasonic vibration on the tool electrode can improve the material removal rate.

### 4.6. Typical Machining Results

After analyzing the influence of ultrasonic amplitude, machining voltage, duty factor, pulse frequency, and feed rate on the slit width, the main machining parameters are optimized to complete the typical structure cutting. After parameter optimization, the micro array slit cutting was carried out on 300 μm thickness glass with a voltage of 34 V, amplitude of 5.25 μm, duty factor of 70%, frequency of 1000 Hz, feed rate of 1 μm/s and spindle speed of 3000 rpm. The complex micro planar coil structure, as shown in Figure 15, has an entrance slit width of 124.1 μm.

Through parameter optimization, in order to verify the machining ability of ultrasonic vibration-assisted wire cutting with helical electrode proposed in this paper, the microcantilever structure with high depth diameter ratio is cut, and 35 μm thickness glass microcantilever structures are fabricated, as shown in Figure 16. The cantilever structure, shown in Figure 16a, has a length of 1500 μm and an aspect ratio of 42, and the disc-free end microcantilever structure, shown in Figure 16b, has a length of 1194 μm and aspect ratio of 13.

## 5. Conclusions

In this paper, the microstructure of high aspect ratio glass cut by ultrasonic vibration-assisted helical electrode WECDM is studied using simulation and experiment. The conclusions are, as follows:(1)The formation process of the gas film on the surface of the helical electrode in ECDM was analyzed, applying ultrasonic vibration to the tool electrode can refine the film thickness, reduce the film thickness, and then reduce the critical voltage of electrochemical discharge, improve the machining quality and enhance the localization and stability of machining. However, if the amplitude is too large, the bubbles are not easy to adhere to, which can result in the non-consistency of the gas film thickness on the electrode surface.(2)The micro helical electrode with a diameter of 100 μm was used for the cutting test. The influence process of the current waveform on the main machining parameters was deeply analyzed. The influence of the main parameters, such as ultrasonic amplitude, machining voltage, duty factor, pulse frequency, and feed rate on the cutting slit width, was analyzed through the experiment. Finally, the combination of parameters is optimized under the influence of the main machining parameters.(3)Using the optimized machining parameters, the glass complex micro planar coil structure with 124.1 μm entrance slit width and the microcantilever beam structure with a high aspect ratio of 42 and 13 are cut by using the optimized machining parameters. The results show that the ultrasonic vibration-assisted WECDM cutting can basically meet the needs of glass and other hard and brittle materials for high aspect ratio microstructure machining.

## Figures and Tables

**Figure 1 micromachines-12-00125-f001:**
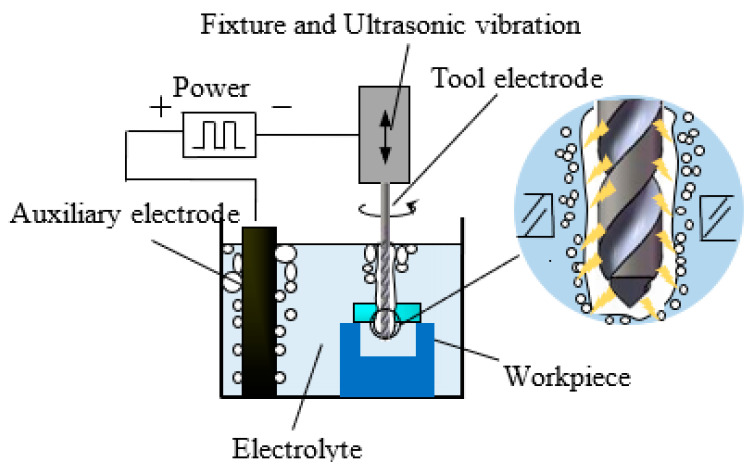
Schematic diagram of electrochemical discharge machining (ECDM).

**Figure 2 micromachines-12-00125-f002:**
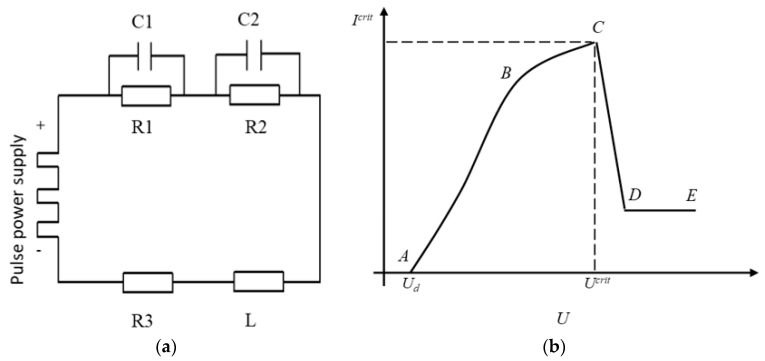
Equivalent circuit diagram and voltage current characteristic curve of ECDM. (**a**) Equivalent circuit diagram. (**b**) Voltage current characteristic curve of ECDM.

**Figure 3 micromachines-12-00125-f003:**
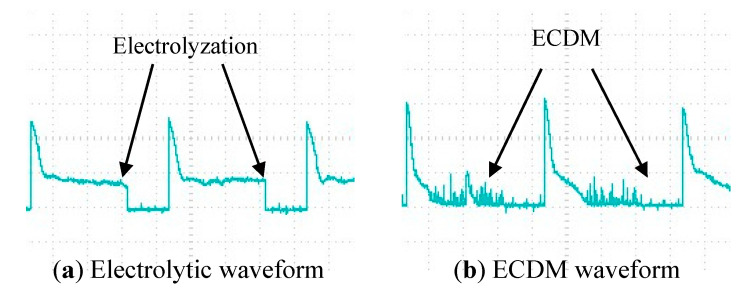
Current waveform from electrolytic jump to ECDM.

**Figure 4 micromachines-12-00125-f004:**
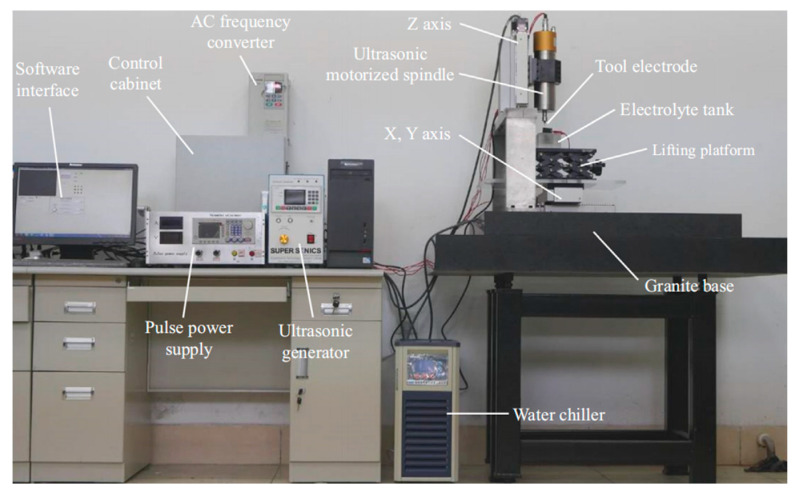
Picture of the physical experimental platform.

**Figure 5 micromachines-12-00125-f005:**
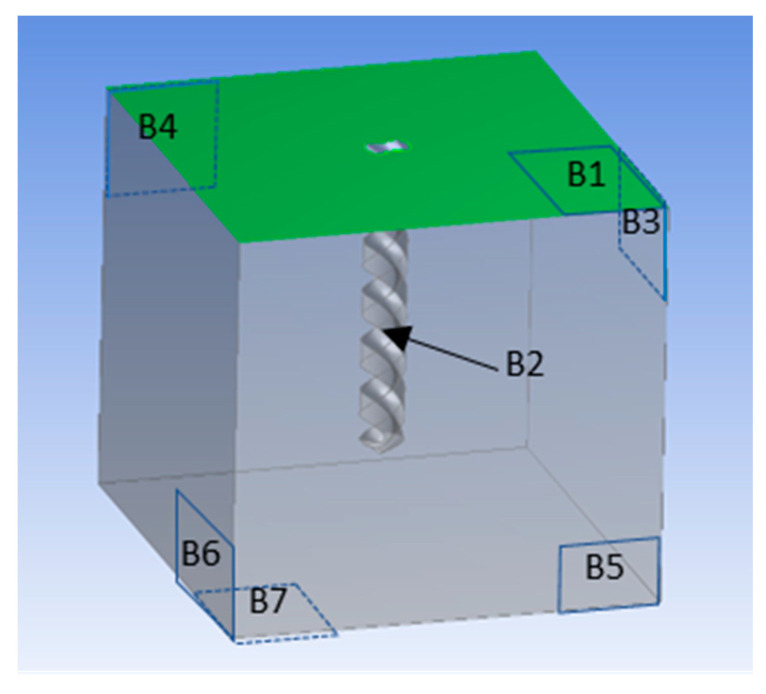
Geometric simulation model of the effect of ultrasonic vibration on gas film.

**Figure 6 micromachines-12-00125-f006:**
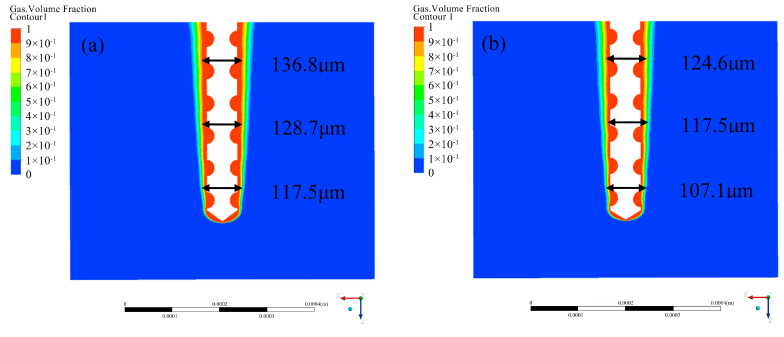
Effect of ultrasonic vibration on gas film. (**a**) Without ultrasonic vibration assistance. (**b**) With ultrasonic vibration assistance.

**Figure 7 micromachines-12-00125-f007:**
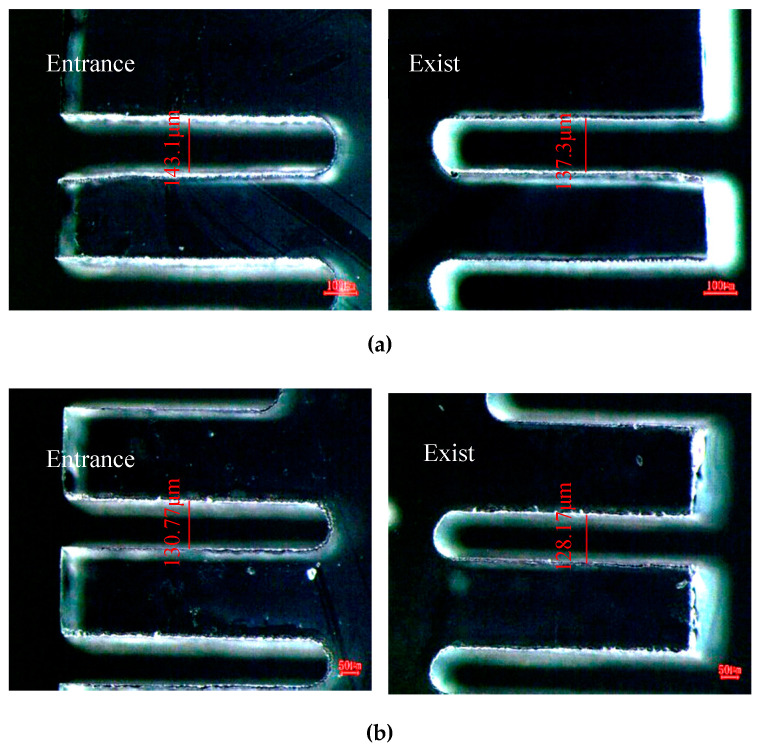
Comparison of slits with or without ultrasonic vibration. (**a**) Without ultrasonic vibration assistance. (**b**) With ultrasonic vibration assistance.

**Figure 8 micromachines-12-00125-f008:**
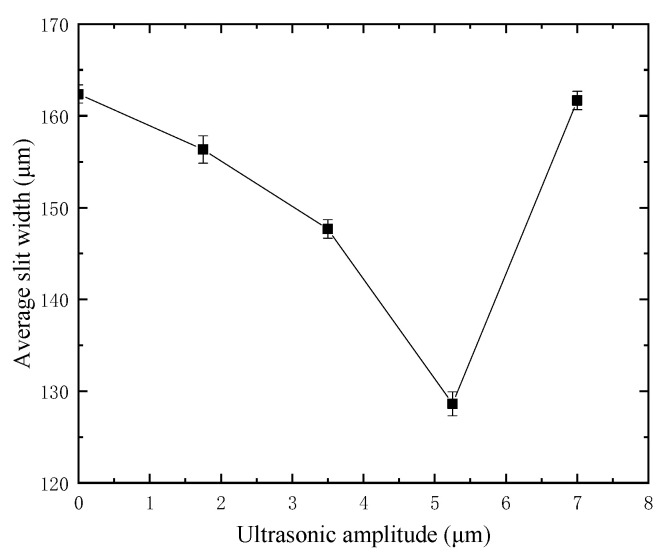
Slit width under different amplitude.

**Figure 9 micromachines-12-00125-f009:**
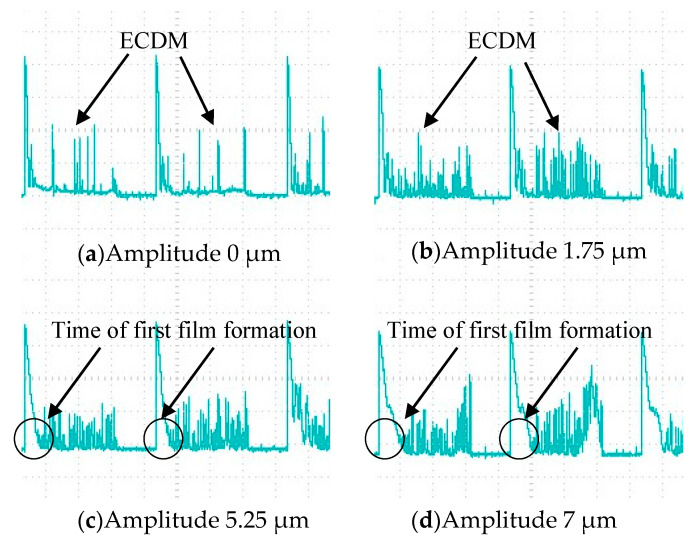
Current waveform under different amplitude.

**Figure 10 micromachines-12-00125-f010:**
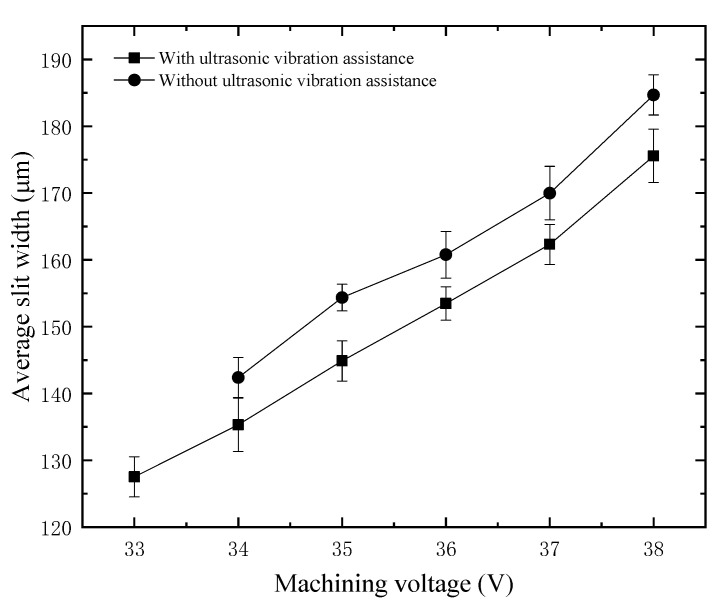
Slit width under different machining voltages.

**Figure 11 micromachines-12-00125-f011:**
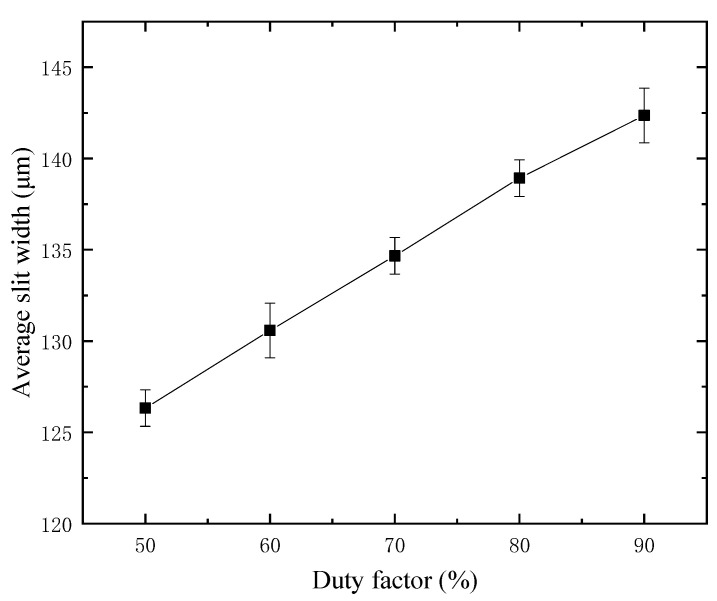
Slit width under different duty factors.

**Figure 12 micromachines-12-00125-f012:**
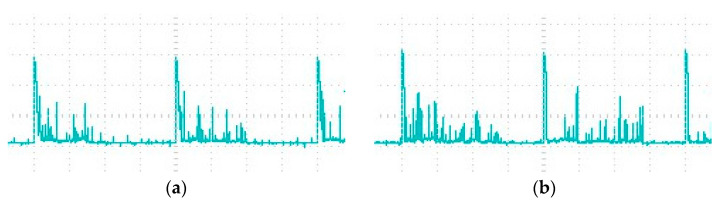
Current waveforms under different duty factors. (**a**) Duty factor 50%. (**b**) Duty factor 70%.

**Figure 13 micromachines-12-00125-f013:**
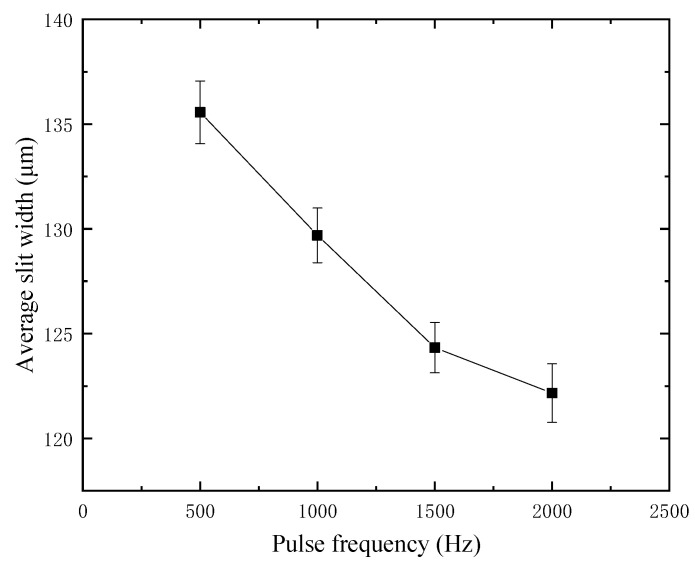
Slit width at different pulse frequencies.

**Figure 14 micromachines-12-00125-f014:**
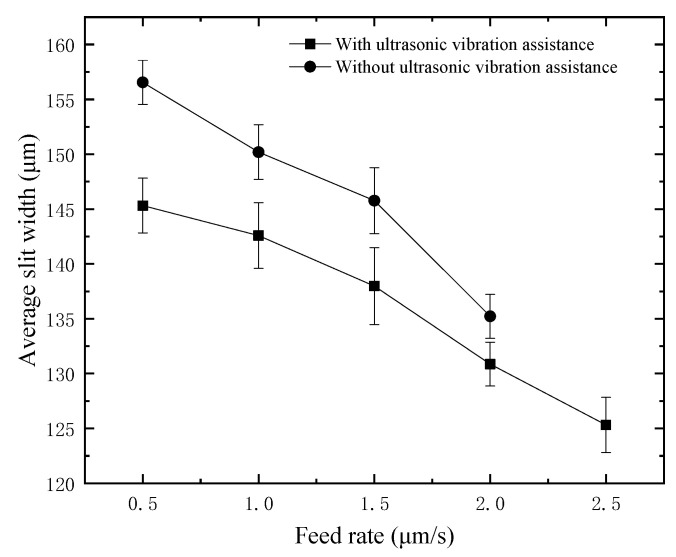
Slit width at different feed rates.

**Figure 15 micromachines-12-00125-f015:**
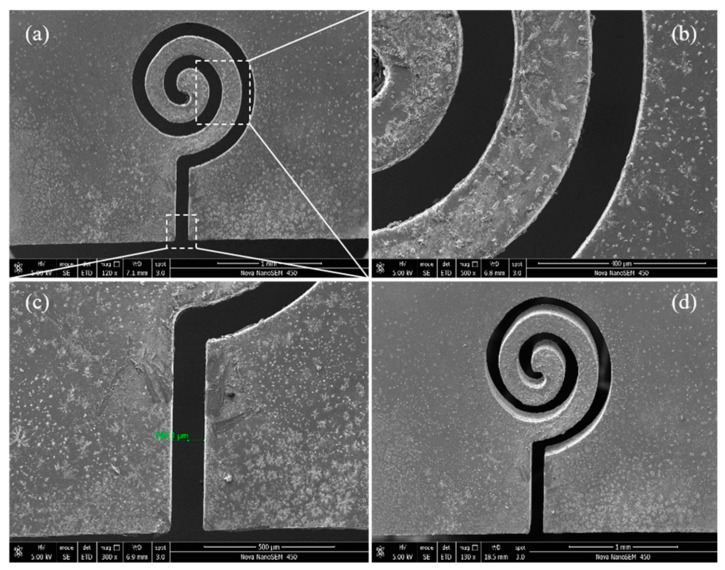
Micro planar coil structure and entrance slit width. (**a**) Front view; (**b**) Enlarged view of spiral part; (**c**) Enlarged view of entrance; (**d**) Rear view.

**Figure 16 micromachines-12-00125-f016:**
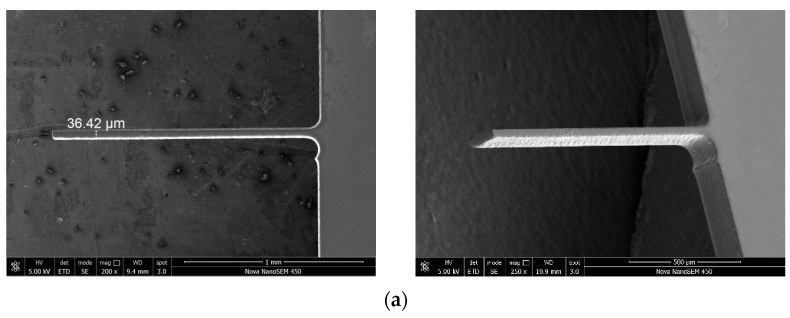
Structure of typical glass microcantilever. (**a**) Structure of glass columnar microcantilever. (**b**) Structure of glass disc-free end microcantilever.

**Table 1 micromachines-12-00125-t001:** Parameter setting of wire electrochemical discharge machining (WECDM).

Machining Parameters	Value Range
Spindle speed	3000 r/min
Electrolyte concentrations	3 mol/L
Workpiece material	Quartz glass
Workpiece thickness	0.3 mm and 1.08 mm
Ultrasonic amplitude	0–7 μm
Pulse voltage	33–38 V
Duty factor	50–90%
Frequency	500–2500 Hz
Feed rate	0.5–2.5 μm/s

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
