# Peer review of "Experimental Study on Ultrasonic Vibration-Assisted WECDM of Glass Microstructures with a High Aspect Ratio"

_micromachines, 2021, doi:10.3390/mi12020125_

Round 1

Reviewer 1 Report

The paper presents interesting process and its application. However before publication Authors should take into account following comments and address questions:

The description of symbols R1, R2 .. etc should be added to caption in figure 2.

In this process electrolysis do not take part in machining mechanism so the sentence It can be seen from Figure 2(b) when the voltage exceeds the critical voltage, the machining can change from electrolysis to ECDM is wrong and should  be corrected (page 4, line 127-128).

The machining principle and machining set-up should be described in separate subparagraphs.

Please include units of vibration frequency (page 6, line 155)

Term cutting speed in not correct for this process.

How you measure and control vibration amplitude.

Page 7, line 202: In the parameter setting, the diameter of hydrogen bubble is set as 2μm, and the average outlet velocity is 0.0005m/s according to the experimental conditions – which experiments, more details are necessary, what about hydrogen bubbles flow rate? How the helical electrode tool vibrates (which axis), what about rotation? What about the workpiece in numerical model? How it influences these results? – above mentioned comments points that the numerical model is poor and needs more detailed results, explanation, discussion of assumptions and linking with real machinign process.

The separate section with research methodology is necessary, what about tool and measurement instruments, critical voltage, current.... . Explain why these machining parameters etc.?

Fig 10 – how you correlate machining voltage with process voltage (it should be much higher)

What about tool wear?

What is physical mechanism of ultrasonic vibration influence on the machining process? Please explain in details.

Reviewer 2 Report

  1. In the paper Authors have presented important experimental research of WECDM process for machining glass microstructures. The problem is very important from scientific and practical poin of view.
  2. Title of the paper iS correct and properly expres the problem taken into account in the paper.
  3. Organization of the paper is correct.
  4. Descriptions of the test stand, measurements methods and equipment are  satisfactory
  5. Literature analysis is  complex and helps to formulate the aim of research and to plan experimental investigations.
  6. General research methodology is correct.
  7. Presented experimental tests results are useful from scientific and technological point of view.
  8.  It would be useful to include  to the paper some information of the workpiece surface quality.
  9. TAKING INTO ACCOUNT ABOVE STSTEMENTS I RECOMEND THE PAPER FOR PUBLICATION.

Reviewer 3 Report

WECDM must be defined in text before acronym is used.

The introduction is very sterile and does not read well. Instead of simply listing references and contributions, the authors should indicate how each of these manuscripts relates to the work that they are producing here.

Lines 127-128. …the machining can change from electrolysis to ECDM. Please provide evidence that machining occurs in the electrolysis (gas evolution) step.

In this paper, the critical voltage from electrolytic jump to ECDM is 23V, and the current waveform is shown. Why is this value assumed?

Line 155 vibration frequency is between 20000 ~ 31000. No units given, I assume Hz.

Line 209. Please use more appropriate time unit, e.g. ms or micro seconds.

It is unclear what the simulation is bringing to the paper, which is a parametric study. In figure 6, it is difficult to determine the difference between the two simulations. Can this be more appropriately quantified?

Figure 6. Units and scale bars cannot be read, please increase magnification of these aspects.

With the help of ultrasonic vibration, the critical value of WECDM can be reduced. The authors need to discuss this and present reasons why. Stating a finding is not enough. This should be done throughout the paper regarding other results as well.

Figures 15 and 16: units and values need to be larger.

Please ensure gaps between values and units, e.g. lines 62, 63, 64, 81, 85, 228 etc. Please carefully check as there are many instances throughout the text.

The text will need to be carefully proof read for English errors.

Round 2

Reviewer 1 Report

The paper was improved so in my opinion can be accepted.